# Exploring Alternative Methods of Visualizing Patient Data

Rabia Tanvir*

University of Toronto

Mark Chignell*

University of Toronto

Deborah Fels‡

Ryerson University

## ABSTRACT

Patient data visualization can help healthcare providers gain an overview of their patient's condition and assist in decision-making about the next steps on management and communication. We explore the acceptance and opinion of five different visualizations that can be used to summarize patient data, including a Text Summary, text and frequency-based Word Cloud, a Bar Graph, a time-based Line Graph and a newly developed Text Graph that combines text and time-based distribution. Results from a user study with 15 professional healthcare providers, 16 first- or second-year medical students, and 17 third or greater year medical students show that most visualizations are useful in extracting patient information and are received positively by the users. In addition, Text Summary and Text Graph are rated to be the most useful visualizations in extracting patient health information.

**Keywords**: Healthcare data visualization, Health information, Patient management.

**Index Terms**: Human-centered computing —Visualizations— Visualization techniques —Graph drawings.

## 1 INTRODUCTION

Health care providers (HCPs) are health professionals whom a person sees when they are in need of medical care or advice. This may include physicians, medical specialists, nurses, etc., [1]. Part of the continuity of this care/advice is documenting the interactions with, and measurements of, the people they see, which becomes the patient's medical record. HCPs use the information found in a patient's medical record to support their decisions on patient care and management [2]. Patient medical records consist of clinical notes and patient data including demographics, laboratory results, radiographic images, problem lists, medication lists, etc., that have been gathered via official requisitions, and use approved, validated measurement techniques [3], [4]. Patient-generated data is a recent trend in medical record data collection, where patients (or their caregivers) record or gather the patient's own health data. The information collected may include health symptoms, lifestyle choices, biometric data, etc. [5]. Specialized technology that can either be provided by an HCP, or a publicly available technology such as a FitBit™, is often used for collecting patient-generated data. However, these data are less formal and possibly less trustworthy because of periodic inaccuracies [6].

* email: rabia.tanvir@mail.utoronto.ca
* email: chignel@mie.utoronto.ca
‡ email: dfels@ryerson.ca

The quantity of patient data collected can be overwhelming to process by HCPs as there can be many individual data items in different styles and formats from a large variety of sources [7]–[9] The information overload increases even further for patients with chronic illnesses as patient data accumulates over time [8], [10]. In addition, sorting through and interpreting patient records can be time consuming and intensive. HCPs are under constant time pressure due to the amount and complexity of patient cases they have under their care. Gathering information, interpreting it, and deciding the next step for their patient must be done as quickly as possible [11], [12]. There is a need to have patient data presented in a concise and summarized manner allowing healthcare providers to efficiently access relevant data and it to manage their patient care [9].

MyHealthMyRecord (MHMR) is designed to allow patients to self-produce brief audio-video recordings of their experiences in-between visits to their healthcare provider [13]. Currently the system lacks a method for visualizing data contained in the recordings. This paper presents an evaluation of five methods for summarizing the patient-generated data from MHMR. The five methods are a Text-Summary, text-based Word Cloud, frequency and time-based graphs, and a newly developed Text Graph that combines text and time-based distribution.

The research questions we aim to answer are: 1) How do healthcare providers interpret visual summaries of patient-generated data presented in the different MHMR visualization formats? and 2) What are the preferences and acceptability of these visualizations for managing patient care? In this paper, we will present the design and implementation of five MHMR visualizations that include a Text Summary, Word Cloud, Bar Graph, Line Graph, Text Graph, and Text Summary. We will then present and discuss our findings from a qualitative evaluation with 15 professional HCPs and 33 medical students. Because these individuals vary in their level of medical training, we want to investigate whether there is a difference between how they perceive the five visualizations and how useful they find them in extracting patient health information. The results and discussion cover the users' rating of different visualizations based on usefulness and their opinions of the visualizations.

## 2 BACKGROUND

Data visualization is the use of graphics to illustrate information [14] that can then be used to support decisions [2]. Visualization enables one to efficiently find trends and outliers within a dataset and understand underlying patterns. It also allows the detection of trends and patterns, which can then be presented and communicated to others so they can understand and make sense of the data as well [15].

### 2.1 Methods of visualizing data

One common method of visualizing data is using a graph that displays a relationship between two or more variables within a dataset [16]. One type of graph that is often used to depict continuous data is a line graph [16]. A line graph connects data points displayed on a two-dimensional scale [16]. An advantage of

a line graph is that it can highlight trend [17], and multiple continuous datasets can be plotted on the same graph for comparison. But, when reading line graphs, individuals spend less time viewing the trends and more time relating different graphical features such as axis and graph titles or data point labels to one another to make sense of the graph [18]. Another example of a common graph is a bar graph, created with the use of vertical or horizontal columns. A bar graph compares a single variable (often the dependent variable) against several variables, and each column represents one group [16].

Word clouds are another method of data visualization. They summarize a body of text by illustrating the words that occur most frequently [19]. A high-frequency word will be shown in the word cloud in large font, and any words mentioned less frequently in comparison will be displayed in a smaller font or not included at all [20]. Text features and word placements are often used to create a word cloud [21]. Text features describe the font colours, font-weight, and font size. Word placement then describes the layout of the word cloud ranging from sorted (e.g., alphabetically), to semantically clustered (e.g., placing all nouns together), and to spatially laid out (i.e., unordered placement of words) [21]. Word clouds are useful for four main activities: searching, browsing, impression forming and recognizing or matching [21]. When an individual uses the word cloud to search, they look for cues such as font size or colour to get a sense of the organization and frequencies of words as proxies for underlying concepts. When users browse a word cloud, they get an overview of the text properties, forming impressions of which concepts are important and inferring information about the text data underlying the world cloud, and in some cases identifying themes that emerge within the dataset [21].

## 2.2 Healthcare data visualization

Some examples of early work in healthcare data visualization include a one-page detailed graphical summary proposed by Powsner and Tufte [22], and Lifelines by Plaisant *et al.*, [23]. The graphical summary could reveal patient condition status to physicians by plotting numerical patient data as a variation of a scatter plot, then adding doctor's notes and relevant medical images on the same page [22]. Lifelines display a patient's history as a timeline where patient visitation dates are on the horizontal axis and information such as patient concerns, diagnosis, medications etc. are presented on the vertical axis as dots or horizontal lines depending on their duration [23]. Both of these studies only have one type of visualization for users. The graphical summary by Powsner and Tufte displays the patient data as a variation of a scatter plot and Lifelines has a single mode of display using a timeline. The MHMR system provides five different visualizations for the users to be able to choose what is best for them in viewing and understanding patient data.

Sultanum *et al.*, [24] developed the Doccurate system to present patient records on a timeline similar to Lifelines and also included a text panel showing associated clinical notes for each patient session. The text panel was added because the clinical text was found to be the physician's primary source of information as they used it to obtain a general understanding of the patient condition, to recall information, and to assist in answering patient-related questions [12]. Additionally, since text-based patient records and annotations have been a dominant part of physicians' practice, text could be useful as a "familiar place to return to" [12, p. 10].

A filtering system in Doccurate allowed users to select a particular medical condition term. As more mentions of these filtering terms occurred in the written notes, the terms appeared in larger fonts on the timeline. A user study with one physician and five residents was conducted to evaluate the system. The participants were asked to compare the patient's information gathered with the system using a set of filter terms they generated themselves, with a set of predefined terms. They attempted to gather information using both sets of terms for two patients. The main findings from this study were that the participants were satisfied with the system but generally had a low level of trust in the automation used in Doccurate, because it made classification errors [24]. Similar to graphical summaries [25] and Lifelines [23], Doccurate only had one method of visualization (a timeline), limiting the user's choice in how they could display the patient data.

The use of different font-sizes to highlight words in Doccurate is similar to MHMR's Word Cloud and Text Graph features but in Doccurate different font-sizes are intended to draw attention to certain visitation dates, and the Word Cloud and Text Graph text are intended to focus on the symptoms themselves.

The Harvest system [26] displayed patient data as a longitudinal timeline, problem cloud, and doctor's notes. The problem clouds (word clouds) demonstrated concepts extracted from the notes, based on the frequency of mention. This system was based on the work of Reichert *et al.*, [8] who asked physicians to create patient record summaries. The aim was to determine which section of the patient record the physicians spent most of their time in creating summaries. Similar to the findings of Sultanum *et al.*, [12], the notes section was used the most by the participants, perhaps indicating that reading text summaries is the easiest way for physicians to achieve an overview. Since users preferred text, the researchers recommended the use of problem clouds and provision of a functionality to view notes.

The Word Cloud visualization in MHMR displays the most frequently mentioned words in the patient videos. In contrast to Harvest, it displays day-to-day variation of particular symptoms. The Line Graph and Text Graph in MHMR illustrate the distribution of a particular symptom over time.

The MHMR system is a mobile application that patients can use to audio-video record their experiences in-between HCP visits. A case study with one patient was conducted to evaluate the use of MHMR, and to explore the topics, and issues that arose during the patient's journey [13]. The patient, who was diagnosed with a chronic disease, used a tablet version of MHMR for three months and documented the frustrations or barriers that were faced. Results from this study showed that the patient was willing and able to create videos about their experience and that there were readily identifiable themes related to health, pain, and accessibility issues. While the task of making video entries may be doable and worthwhile for patients, the information contained in the videos could also be useful to their HCP in understanding events, activities and issues that arise between visits and that may affect the patient's ability to manage their health conditions. However, patients may generate a large number of video materials. Asking HCPs to watch, analyze and understand a large set of videos in the time that is usually allocated to individual patients for an HCP visit is unrealistic and may interfere with the quality of the interaction (e.g., through factors such as reduced eye contact). Thus it was important to organize the large quantity of data in MHMR so that it could be useful to HCPs in managing their patient's care by incorporating the concerns, activities and progress identified in the videos into consultations and decision-making [13].

## 3 METHOD

A user study was designed to evaluate the usability and visualization preferences of MHMR from three different groups of HCPs (first- or second-year medical students, third or greater year medical students, and professional healthcare providers). The study was conducted online with a desktop computer using Zoom

conferencing services [27]. A prototype web application for the visualization aspects of MHMR was developed and deployed to GitHub Pages [28] for the online study. In addition to the five visualizations, the prototype also had a set of six samples of 30-second personal health videos. These videos represented examples of a patient's perspective on pain over a period of time. This study was approved by Ryerson University and the University of Toronto's ethics board.

### 3.1 MHMR Visualizations

Iterative design of the MHMR visualizations was carried out, with three versions being generated over a two-month period based on feedback from the MHMR team. The MHMR team consisted of individuals with experience in computer science, human computer interaction, and 5-10 years of medical experience. Although not part of the system initially, filters/sorting options were added to the system to provide customization, based on feedback from the team. Over the second and third iterations, the correct type (e.g., physical health symptoms, all words, top ten words mentioned, etc.) and wording (e.g., health symptoms vs. physical health symptoms) for the filters were selected.

Currently, there are five visualizations that are used in the data visualization study for MHMR: Word Cloud, Bar Graph, Line graph, Text Graph, and Text Summary. These are made available in a mobile app that simulates the visualization functionality that could be used in the prototype MHMR application. To generate the visualizations, audio from the patient videos was first transcribed into text using IBM Watson's speech to text feature [29]. Common "stop" words such as "the, and, but, how, a, etc." are eliminated from the transcript using a natural language toolkit inside a python script. The remaining words and their frequency of occurrence in the videos are then used for creating the five visualizations. In this section, the examples show the occurrences of words from a sample patient video set.

#### 3.1.1 Word Cloud

The Word Cloud example used in this MHMR data visualization is generated by adding all words extracted from the transcript to an online tool, WordItOut [30] (see Figure 1). This tool allows for the selection of font colour and style that match the other visualizations. Within the MHMR data visualization tool, the user is able to sort and organize the Word Cloud by selecting a minimum number of word occurrences e.g., 5 or more instances, 10 or more instances and more than 30 instances. The user is also able to filter between "All words" and "Physical health symptoms". The "All words" option displays all words that are present in the transcript providing a birds-eye-view of all of the experiences reported by the patient during the recording period. When the word cloud is limited to only "Physical health symptoms", HCP can focus on health-related issues. Words such as "pain" and "swollen" in the "Physical health symptoms" option are extracted from the "All-words" list.

#### 3.1.2 Bar Graph

The Bar Graph example illustrates the ten most frequently mentioned words in the videos and displays them with five sorting and filtering options (see Figure 2). The x-axis represents the words, and the y-axis represents the frequency of the word in the videos. The filters allow users to choose between the top ten words mentioned, physical health symptoms, or the top ten words with additional adjectives or nouns. For example, the word "swollen" from the "Top ten words mentioned" graph becomes "Ankle swollen" in the "More description" graph. The user is able to sort the order of the word frequencies in the graph alphabetically, highest-to-lowest-mention, group by positive or negative

sentiments, and group-by-ranges. Grouping by positive and negative organizes the Bar Graph based on the sentiment of the videos. Colour is used to indicate whether the word has a positive or negative sentiment (orange is used for negative and green for positive). For example, one of the words in the Bar Graph is "walking"; once a user applies the positive/negative sentiment filter, the bar becomes green indicating that this word is associated with positive sentiments in the videos. The group-by-ranges filter groups words together based on frequency size. For example, all words between 10-29 mentions are grouped together, 30-49 in a second grouping and finally more than 50 in a third. A dotted rectangle surrounding the "Pain" bar represents a button that takes the user to the associated Text Graph of a certain word. Since there is only one Text Graph in the prototype and it represents pain, this Bar Graph only has one button.

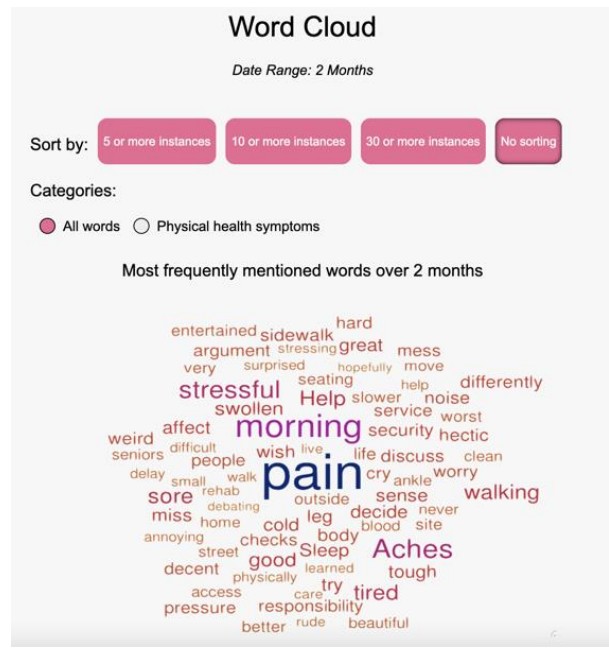

Figure 1: Word Cloud example used in the MHMR data visualization study.

#### 3.1.3 Line Graph

The Line Graph example (see Figure 3) represents the data as occurrences of words over specific time intervals. The example in Figure 3 shows the number of negative mentions of pain per day over a two-month recording period. The x-axis represents the time interval in days over the two-month period, and the y-axis represents the number of times the word pain is mentioned with negative sentiment in the videos for a particular day. The clear or black-filled circles plotted at the 0 points are either when there was no video created that day or no videos where there were negative pain words respectively (the legend below the graph indicates the meaning of the clear and black-filled circle). The two points surrounded by the dotted rectangle indicate a button that when clicked will take users to the videos created for that particular day.

#### 3.1.4 Text Graph

The Text Graph (see Figure 4) adds text markers to maxima and minima points on the time-based Line Graph. In the example shown in Figure 4, the maximal and minimal of the graph are labelled either "More pain" or "Less pain" indicating whether the video(s) for that day mentioned pain in a positive or negative manner. The

font-sizes of the text vary with the frequency of mentions as in a Word Cloud. The graph also has filters for users to choose to show positive only points, negative only points, or both. In addition, users are able to toggle between a coloured and black/white representation. We wanted to determine whether colour could help people interpret a text graph that was populated with a large variety of information displayed (e.g., lines, labels/text, points, and button indicators). When there is a day where no video is created, or there is no mention of pain, the graph shows clear or black filled-in circles, respectively. Points on the graph shown in Figure 4 with a dotted rectangle are buttons that when selected lead users to the videos created for that day, thus offering users a drill-down option.

high-frequency words. It is created using the data from the transcripts. The purpose of the summary is to briefly describe the main patient experiences over the duration of the entire video set. For example, "aches" and "pain" were mentioned often in conjunction with the word "morning". The Text Summary then shows "The patient complained of pain and aches multiple times...On most occasions, complaints of pain and aches were mentioned with "morning"". The user is able to toggle between a general summary and a quantitative summary. The quantitative summary displays the number of videos that mention a certain word. For example, "59/72 videos mentioned "pain" with a total account of 90 mentions".

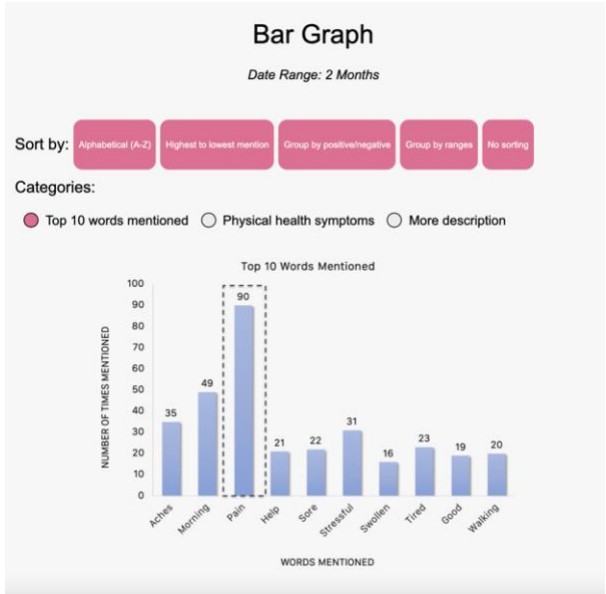

Figure 2: Bar Graph example used in the MHMR data visualization study.

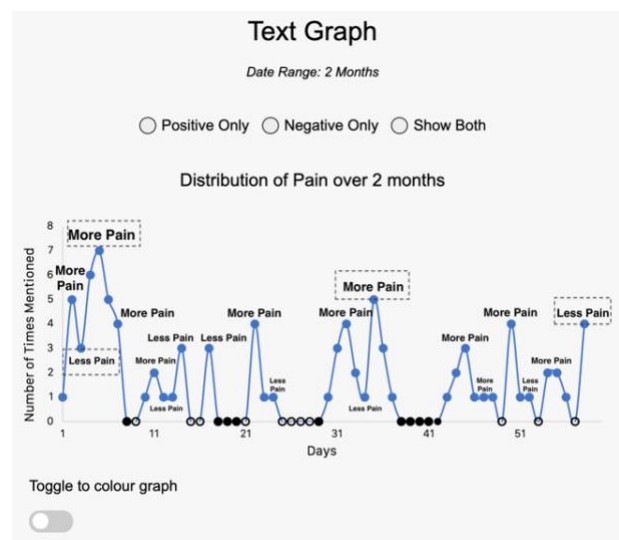

Figure 4: Text Graph example used in the MHMR data visualization study.

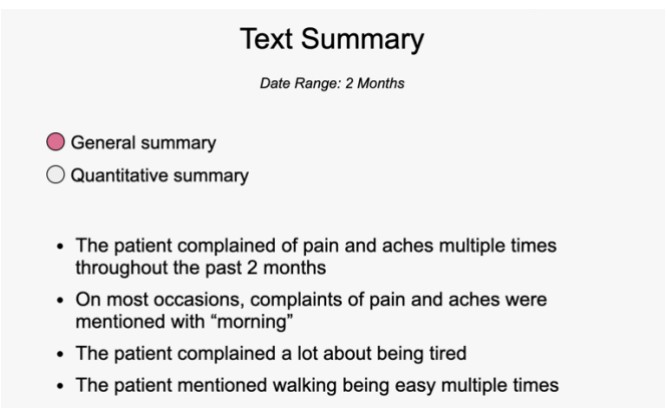

Figure 5: Text Summary example used in the MHMR data visualization study.

### 3.2 Participants

A total of 48 healthcare providers and individuals in medical school (20 males, 27 females, 1 did not answer) were recruited for the user study. The three groups were: students currently in their first or second year of medical school (16 in total), students in their third or greater year of medical school (17 in total), and finally healthcare providers with two or more years of work experience in the healthcare industry (15 in total). The healthcare providers were from a range of disciplines including registered nurses, graduate nursing students, medical residents, a general practitioner, a

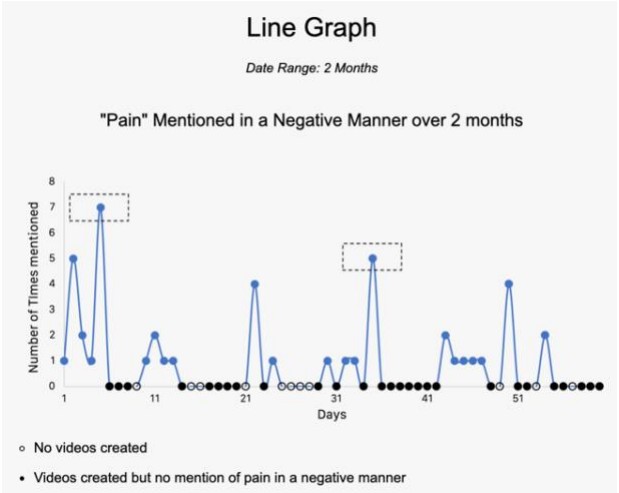

Figure 3: Line Graph example used in the MHMR data visualization study.

### 3.1.5 Text Summary

The Text Summary example (see Figure 5) is a general summary of the videos presented as text-based bullet points created from

nutritionist, and a behaviour therapist. These three groups were chosen because they vary in their experience of medical training. Previous studies [8], [12] have shown that professional HCPs like to use clinical notes to support their decisions, but we wanted to see if the amount of training can play a role in how HCP like their patient data presented. The first group were freshly starting medical school and lacked exposure to traditional patient data summarizations such as clinical notes. The second group of students had slightly more practical medical training, perhaps including clinical rounds, so they had more experience in handling patient data than the first group. Finally, the third group were individuals working in the field and had the most exposure and experience in handling patient data. Age, gender and years of experience were collected to ensure that there was a representative sample of the target populations (see Figure 7). Participants varied in age between 18 and 41 years with the majority of participants (31 of 48) aged between 18 and 22 years. All participants were given a small token of appreciation for their participation in the study.

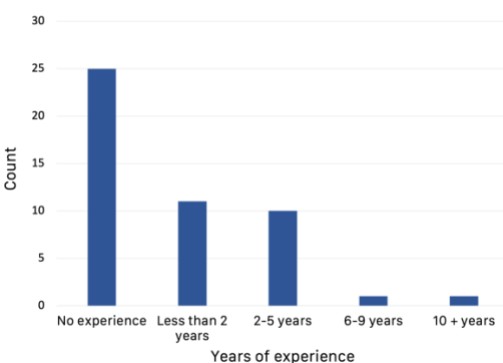

Figure 6: Distribution of medical experience of participants.

### 3.3 Study design

Each study lasted around 90 minutes and began with a pre-study questionnaire that gathered demographic data as well as comments on the healthcare provider's/medical student's current routine of practice. This was followed by a short training period where the user was introduced to the visualization system as well as a patient vignette and scenario. The vignette explained the patient's condition and their experiences to the user, whereas the scenario set the tone for the user study explaining what exactly the user will be required to do. Participants were then invited to complete ten tasks while thinking aloud followed by a short semi-structured interview that gathered their opinion of the visualizations. Three versions of the ten user tasks were created, and the participant was randomly assigned a version. Each version had the same user tasks but in a different order to eliminate sequence bias. The tasks were designed so that the user had to use the application and the visualizations to answer the questions, but users could use the application as they preferred, even if it meant that they only used one or two of the visualizations throughout the study. However, for each task one or two visualizations were more appropriate for answering the question. For example, one task was "How did the patient's pain vary over the course of 2 months?". Since the task asks a time-based question, the Line Graph or Text Graph would be more appropriate to use. The users were free to skip any question they did not feel comfortable or had trouble answering. Notes were also generated on what visualizations the participants used the most for the study tasks, and which ones they were struggling to understand. The study ended with a post-study questionnaire that allowed participants to rate the system usability, reflect on their experiences, discuss what they liked/disliked about the system, and make recommendations. In this paper, the results of the post-study questionnaire and observational notes are reported.

### 3.4 Data collection

The post-study questionnaires consisted of ten System Usability Scale [31] questions, two questions that allowed participants to choose which visualization(s) they liked the most and which they liked the least, and one 4-point rating questions on the perceived usefulness of each visualization. Usefulness was rated with four possible responses: Very useful, Useful, Somewhat Useful, Not useful at all. There were also five open-ended questions that allowed participants to use freeform text to write about their opinion and interest in working with the system and its visualizations.

### 3.5 Data analysis

The questionnaire responses were analyzed using non-parametric statistical methods. A Kruskal-Wallis non-parametric analysis of variance was used to determine whether there were significant differences between the three participant groups for usefulness and ratings of each visualization. Then, Friedman One-Way Repeated Measure Analysis of Variance by Ranks was used to test whether there was a significant difference in ratings of all visualizations. A Wilcoxon Signed Rank Test was then used to check where these differences occur. Finally, the strength of association between the ratings of different visualization was assessed using Kendall's tau.

### 4 RESULTS

The mean SUS score was 74.90 (SD=13.80). According to Bangor *et al.*, [32] a score above 68 demonstrates average usability obtained across a range of studies. In addition, 60.4% of the participants were very likely (rating of 5 on a 1-5 scale) to recommend this system to a friend or colleague.

### 4.1 Statistical analysis

The Shapiro-Wilk test was used to check for statistical normality of the usefulness ratings. The test results indicated that the ratings distribution departed significantly from normality ($p < 0.05$). A Pearson chi-square test of independence was used to determine the significance of usefulness ratings for each visualization. The test results were statistically significant ($p < 0.05$). A Kruskal-Wallis test showed no significant differences between groups and their rating of each visualization ($p > 0.017$; after Bonferroni adjustment). However, there was a significant difference in the overall rating of each visualization (Friedman Test, $p < 0.017$) and also some pairs of visualizations (see Table 1). In addition, a contingency table analysis was performed between each visualization, but the results were not statistically significant at the .017 level.

### 4.2 Frequency responses

Figures 7 and 8 show which visualizations were preferred or disliked for the different participant groups. Overall, the most preferred visualizations were Text Summary and Text Graph. The most disliked graph was the Line Graph. Figure 9 illustrates the frequency of usefulness ratings for each visualization. Text summary and Text Graph were also rated the most useful visualizations (rating of very useful or 4/4), and the Word Cloud and Bar Graph are rated as useful (rating of 3/4). The Line Graph was rated as somewhat useful (rating of 2/4) by 18 participants.

Table 1: Wilcoxon signed-rank test results on pairs of visualization (TG=Text Graph, LG=Line Graph, WC=Word Cloud, TS=Text Summary). The remaining pairs did not yield a significant difference ($p > 0.017$).

|  | TG-LG | TG-WC | TS-BG | TS-LG | TS-WC |
|---|---|---|---|---|---|
| Wilcoxon Signed Ranks Test | Z=-2.936 $p = 0.003$ | Z=-2.566 $p = 0.010$ | Z=-2.985 $p = 0.003$ | Z=-3.769 $p < 0.017$ | Z=-3.639 $p < 0.017$ |

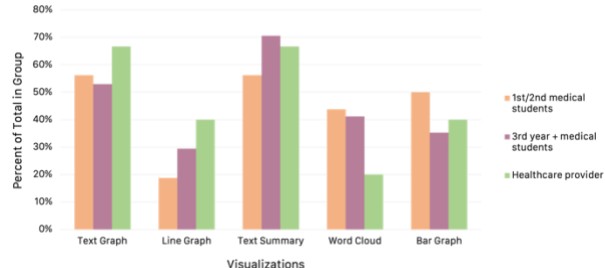

Figure 7: Presentation of most liked visualizations by each group. The data has been normalized by the number of participants (16 first/second year medical students, 17 third year + medical students, 15 professional HCP).

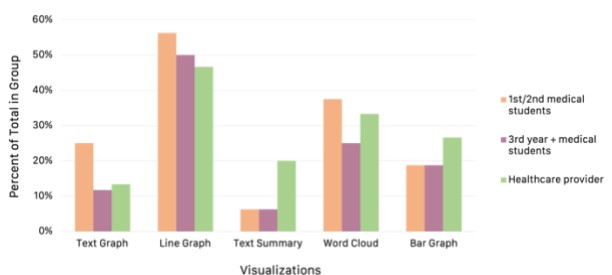

Figure 8: Presentation of most disliked visualizations by each group. The data has been normalized by the number of participants (16 first/second year medical students, 17 third year + medical students, 15 professional HCP).

### 4.3 Written responses

All 48 participants commented on their experience and opinion of the visualizations, and the application. Most participants had positive reviews of MHMR and its visualization techniques. Participants mentioned that the Text Graph or Text Summary were the most useful visualization to work with, e.g., "Text Graph, Text Summary [are the most useful] because they give a better and a quick picture [answering] my questions" (P18). Some participants mentioned that what they liked least about the system was the Line Graph e.g., P6 wrote "Line graphs, I think it is a lot more time consuming and comparatively less helpful than the other techniques." P46 also mentioned that "I think some of the visualizations were redundant (the text graph was a better version of the line graph)". With respect to design and layout, some participants had negative comments on the x-axis labels on the Line or Text Graph because they found them "unclear" (P28) or did not understand the difference between the open and closed circle (P17).

Participants also stated that they would be willing to use this system in their practice especially when monitoring a patient's condition over time, or prescribing medication. For example, P13 wrote, "I do weekly check-ins with my clients, it would help me to see their progress as well as help me to pinpoint where changes in their nutritional and exercise plans need to be made." P31 stated, "As a nurse, you can understand at what time of the day the [patient] experiences more pain, and you can advocate for the [patient] to get pain meds prescribed at certain times of the day." But there were concerns on how compliant patients would be with using MHMR, and how they would be encouraged to record their symptoms as often as possible (P45).

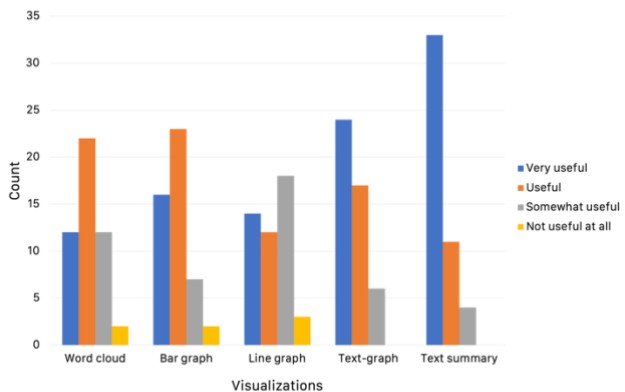

Figure 9: Presentation of usefulness rating for each visualization.

## 5 DISCUSSION

This study evaluated opinions about, and acceptance of, the data visualizations presented in the MHMR application. In the ratings of preference, usefulness (Figures 7-9) and written responses, the Text Summary and Text Graph visualizations tended to be more preferred.

There was a significant difference in rating of Text Summary compared to the Word Cloud, Bar Graph, and Line Graph. In their research, Sultanum *et al.*, [12] concluded that text is a familiar method of generating and consuming information about patients for physicians (or HCP). Thus, it is not surprising that the Text Summary was preferred, since it resembled the type of information provided in clinical notes.

Given that word clouds are used for searching, browsing, impression forming and recognizing or matching [21] it was anticipated that participants would find them useful in finding patterns for matching specific health conditions. For some participants this was the case, and they found the Word Cloud to be useful for this purpose, either alone or along with Text Graph or Text Summary (e.g., P4, P14, P24, and P34). However, for others there was too much disorganized information contained in the Word Cloud, e.g., one participant (P12) stated: "[Word Cloud] appeared scattered and packed", and thus they did not find it as useful as the Text Summary and Text Graph in gaining information about the patient's condition.

The Bar Graph was the only visualization that had a good usefulness rating (Figure 9) and no negative comments associated with it. This could be due to familiarity with its style and the information it conveyed. For example, P37 liked the Bar Graph because it displayed "the most words said and how many times the patient actually said them." Plus, the Bar Graph has a filter that allows participants to sort from highest to lowest frequency words. The usage of this feature was repeatedly observed in the user

studies and as an example, the Bar Graph was liked by P11 because "it gives you [the] highest symptoms experienced vs. lowest".

The Line Graph was collectively the least favourite visualization among the participants. This could be because it requires attention to understand the trend and to investigate each point, or because participants had less experience with interpreting line graphs. For example, P16 said "I think there should be more detail to the line graph because I couldn't understand that graph much", and P30 said "I think the line graph was a bit difficult to read".

The Text Graph provided more or less the same information as the Line Graph but combined text labels with the graphical, time-based representation of the data. The Text Graph was rated more highly than the Line Graph and there was a significant difference between their ratings. 11 participants that rated Text Graph as a 4/4 (Very useful) rated the Line Graph as a 2/4 (Somewhat useful). Carpenter and Shah [18] found that, when viewing line graphs, individuals spent more time relating the different graphical features axis and data point labels to make sense of the data and less time viewing the pattern or trends. The Text Graph may have helped participants make sense of the data because it highlights the important information for them, and so they can focus more on understanding the overall pattern. In addition, The Text Graph allowed users to drill down to extract more detailed information by viewing specific videos related to those data points. This may have provided the additional detail as suggested by P16 or it may have added a sufficient amount of text to take advantage of the familiarity of text favoured in the Text Summary. For example, P32 said "in my opinion, the most useful technique is text-graph [because] it shows day to day variation of patient symptoms...it will help me get a better understanding of my patient [to] evaluate necessary management".

Researchers have found that healthcare providers prefer the notes section in a patient record [8], [12]. The Text Summary in MHMR was similar to clinical notes so it was expected that most participants would show a preference for the Text Summary and would find it useful. Nonetheless, participants also saw benefits of other visualizations, particularly the Text Graph, and formed mainly positive opinions of them. The Text Graph was newly developed for this research and was new to all participants. The Text Graph was designed to exploit the preference for notes and the benefits of visually representing patterns over time as a line graph. The Text Graph and Text Summary had very similar ratings as well. 16 participants rated both visualizations as 4/4 (Very useful) and 12 rated both graphs as 3/4 (Useful) indicating that both visualizations were useful in extracting information about patients' status and conditions. The Word Cloud was also text-based but there was a significant difference between its rating compared to Text Graph and Text Summary. It was mainly rated to be "useful" (3/4) rather than "very useful" (4/4) like the Text Summary and Text Graph. This could be because the Word Cloud was something new for them and they were more comfortable with the Text Summary and Text Graph but were open to trying the Word Cloud as well.

In terms of design, the Text Graph, Text Summary, Word Cloud, and Bar Graph seem to be acceptable ways of visualizing qualitative patient-generated data for MHMR. The Line Graph is not as useful and can be given less importance in the MHMR implementation since it does not clearly convey patient information. In addition, the use of colours on visualizations allows users to distinguish between different aspects and it is beneficial to use them to communicate the results of different filters to the end user.

## 5.1 Limitations

This study evaluated the acceptance and opinion of data visualizations presented in the MHMR application. The statistical analysis of post-study questionnaires showed no significant differences between the groups (HCP, first or second-year medical students, third-year or greater medical students). One of the reasons for this could be that there is not enough data to work with because although data of 48 participants were analyzed, there were only around 16 participants in each group.

### 5.1.1 Demographics

There were no differences in the usefulness ratings between groups, and we suggest that a larger sample may elicit differences. The HCP were mostly nurses who may have different experiences than doctors or other types of HCP. Future studies should incorporate a more diverse set of participants varying in roles. The students recruited were mainly from the same geographic location and thus diverse geographic samples, and the impact of different training regimes between different jurisdictions should also be studied.

### 5.1.2 Online study

One technical limitation was that this was an online study. Technical difficulties such as Internet issues with several participants slowed the process of viewing and interpreting visualizations causing frustration and impatience by participants. This may have impacted their views and they may have been distracted by the technical issues. Another limitation of the online study was that the MHMR was intended as a mobile application, but this study used a responsive web application that mimics the user interface of a mobile application. Users saw a simulation of a mobile screen on a desktop (height of the interface was 980px and the width was set to 50% of the displaying screen) instead of a display on an actual mobile device. Because users were using screens with different aspect ratios, the user interface could appear wider than intended or disproportional depending on the size of the screen, and sometimes the participants were not able to see the entire application at once and had to scroll up or down. This may have had an impact on the participants' view of the application and the visualizations as some information may have been hidden or not clearly visible on their screen. In addition, the MHMR system was designed for a touch screen but the participants used a mouse in the study as they were working on their desktops. This may have affected the ease of use, for example some elements on the graphs may have been easier to touch rather than click using a mouse.

### 5.1.3 Visualizations

Another limitation of the study was the number of visualizations presented. Our study presented five visualizations, however, there are a number of other ways to present data. Other common graphs include scatter plots, pie charts, histograms, etc. In addition, there are a number of ways to add or remove details from graphs to create variation. The Text Graph added text labels on top of a time-based line graph for more information but even simply removing data labels from the Bar Graph can potentially create a difference in understanding of the graph. Future studies should incorporate different types of visualization techniques and assess how different graphical features play a role in the understanding of the data.

## 6 CONCLUSION

The design and implementation of five visualizations depicting patient-generated data were presented in this paper. Participant data from three different groups representing a spectrum of healthcare providers in terms of their education and experience was evaluated for comparison and correlation. Quantitatively, there was no

difference between the groups and their preference and opinion of the visualizations, but there were overall differences in ratings and preferences towards the different visualizations. The results showed positive attitudes towards some of the visualizations in addition to the Text Summary, particularly the Text Graph. The Text Summary was similar to the notes section in a patient record so, as anticipated it was the most preferred and was rated to be the most useful by the users. However, the Text Graph, despite being something the users have not seen before, was as useful as the Text Summary. Many participants were also interested in using this application in their future clinical practice. Future work needs to incorporate a larger sample size and a diverse group of participants. The visualizations also need to be automated and tested for their accuracy in depicting the correct words spoken by the patient and associating the correct sentiment to those words.

### ACKNOWLEDGMENT

Funding for this project was generously provided NSERC, project #355320-07. The authors also thank the participants for their time and generosity as well as the MHMR team for their time and effort on this project.

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
