# OpenReview forum: "Exploring Alternate Methods of Visualizing Patient Data"
_graphicsinterface.org/Graphics_Interface/2021/Conference — GI 2021_

### Official Review · AnonReviewer2 · 2021-01-14
**Exploring Alternate Methods of Visualizing Patient Data**

**Rating:** 5
**Confidence:** 4

**Review:**

This paper explores five different graphical and text-based visualizations for summarizing patient-generated data in the MyHealthMyRecord system. The authors conducted a remote user study with 48 participants belonging to three different groups (e.g., professional healthcare providers, first/second-year medical students, and third or greater year medical students) to evaluate the usability and perceptions of each of the visualizations. The key results from the statistical analysis and qualitative data showed that the participants preferred the Text Summary and Text Graph visualizations and that there was no significant correlation between medical expertise and preferences for visualizations across the different groups.

Overall, this is an interesting paper and I was impressed to see the scale of the study and the effort that went into recruiting medical professionals of varying expertise. Recruiting and working with such populations can be difficult due to their busy schedules, so it is great to see that the authors were able to include a large number of these participants in their evaluation.

Having said that, this paper was a bit difficult to read and the core contribution of this work to HCI and/or Visualization is unclear (and hasn’t been explicitly stated anywhere in the paper). One key takeaway is that the medical professionals in the study preferred text-based visualizations for understanding patient-generated data. But, to the extent that this finding is novel and interesting is unclear to me. There are some other issues with the paper in its current form as well, as noted below.

The paper mentions the design and implementation of five different visualizations, but it is difficult to tease out the design phase of these visualizations and whether there was consideration of any particular design goals. Furthermore, the use of different visual variables such as colour to represent positive and negative sentiments is not clear and inconsistently represented in the figures (e.g. in the bar graph). The visualization decisions taken in terms of representation, presentation and interaction choices are not clear nor convincing.

Furthermore, it was not clear if a certain visualization intended to solve a certain purpose in the user study, which was a bit of a disconnect from the earlier description in the paper. It seems that the ratings were dependent on the particular tasks that were prescribed. Since different tasks could be intended for extracting different information, it may be that some of the tasks could be biased for text cloud usage—it would be helpful if the authors could clarify this point in their revisions. Moreover, the reasons for preferences were not convincing as in some cases the usefulness ratings for a visualization did not align with user’s preferences.

Minor: Figures 6-7 take up a lot of space and do not convey much new information that cannot be easily summarized in the paper in a couple of sentences.  In Figure 8, it is not clear what “years of experience” refers to? In understanding patient data in general or in using visualizations?

Overall, this paper has a solid motivation and some interesting results, but it appears to be too preliminary for publication. As the authors acknowledge in their discussion, there are some significant limitations and threats to validity of the results, making it difficult to assess whether or not the paper is making a novel contribution to the fields of HCI and/or Visualization.

---

### Official Review · AnonReviewer1 · 2021-01-14
**This is a user study paper in which test 5 different visualization methods for patient's speed words extracted from videos. The experiments are designed well and could support the conclusion based on the user study. These user studies is useful to further visualization design for this specific visualization.**

**Rating:** 6
**Confidence:** 3

**Review:**

This paper is pretty clear for me to read. The testing data, user study experiments and testers is very clear introduced in this paper.

This paper's topic is a bit away from by own research background. I could not evaluate whether the relaed works section is enough or not. Based on the current related works, many existing works are mentioned. It seems that the existing works did not explorate so many visualization in a single user study. And some works did not visualize the data as the method used in this paper.

The main drawback of this paper is the limated testing data. There are many kinds of patient data, not only the words extected from videos. During health care process, the patients will do many test about many different features about themself. It's more interesting to test the 5 visualization methods for those data.

The five visualization methods are clear introduced in this paper. However, the author did not try to present a specific novel visualization methods for the specific data. Then try to prove that the proposed methods are better than the existing methods.

---

### Official Review · AnonReviewer3 · 2021-01-14
**visualizing patient data**

**Rating:** 6
**Confidence:** 4

**Review:**

The presented paper reports on a user study with 48 participants (2/3 students in the healthcare field, 1/3 practitioners) where they evaluated the usefulness of 5 representations of patient data.  Given a scenario with fake data, and 10 tasks (questions relating to the data), participants interacted with the system, then provided their opinion of the different representation options.  Participants most preferred the textual summary, followed by text + timeline, ranking the other types of graphs more poorly.

QUALITY:   The paper appears sound and presents sufficient detail to understand the work.  It is clearly organized, well-written, and is generally well done.

One of the limitations of the paper, which is buried on the last page is the issue of the presentation format.  The system was meant to be a mobile app, but was instead displayed on a laptop/desktop screen, which stretched things and sometimes forced them to scroll.  I sympathize because I assume the authors had to switch to an online study due to the pandemic -- but in a visualization study, this seems like a pretty big deal (and may have contributed to participants' favouring the text representation).  Could there at least be a bit more discussion about the implications of this?

CLARITY:  This may be due to trying to anonymize the paper, but the relationship between MHMR and the visualizations was a bit difficult to untangle (at various points I thought MHMR was the novel system being proposed in this paper, then I thought it was an existing system by other authors just mentioned as background, then I thought it belonged to the authors but was being extended).  Clarifying the relationship that would help.

It also wasn't clear whether the implementations of the visualizations was being claimed as a contribution.  If so, more detail about how the author went from the "video data" to data that can be displayed in the visualizations is needed. For example, how were the "physical health symptoms" extracted for filtering?

For the line graphs, how were participants supposed to figure out the meaning of the filled circles vs empty circles? There doesn't seem to be any legend.

On a related note:  During the sessions, how much instruction were participants given about the graphs and how to interpret them?  Also, were they told to use specific graphs for specific tasks?  Or could they use whatever they wanted?  If it was open-ended, how did the authors ensure that the participants interacted with each type of graph sufficiently to actually have an opinion about them?  (Could they have defaulted to text because that's just the first one they understood and they didn't really bother with the others?)

There are no details about how the qualitative analysis is complete.  It seems to have been fairly superficial thus far, but some detail about the analysis process would be useful.

Minor:  Table 1 needs a better caption actually describing what is being compared as opposed to just the name of the test.

Minor:  I think there's a missing word or something in this sentence -- "The quantitative summary displays the  number data in the text."


ORIGINALITY:  The study presents visualizations that extend what appears to be an existing system created by the authors.  The five representations are sufficiently different from each other and represent and interesting comparison.  I think they were somewhat simplistic and agree with the authors that other options are possible, in particular if interactivity is incorporated.

SIGNIFICANCE:  The paper makes a reasonable contribution and should probably be published.  Given the format of their study and the length of the sessions, it seems like there should be richer data available for more in-depth analysis.  I was hoping for a bit more depth, especially in the reporting of the qualitative results and in the Discussion section of the paper.

The conclusions seem a bit strong given what was evaluated and the results.  Saying that participants had positive attitudes towards visualizations when they most preferred the text representation seems like a stretch.   Basically the text representation acted almost as a control condition, and none of the others did better than it.     I still think it's an interesting study, but caution should be used to not over-claim.

Overall, I think there are lots of tweaks that could/should be done to the paper, but it's likely above the bar.

---

### Meta-Review · Area_Chair1 · 2021-01-16

**Recommendation:** Accept
**Confidence:** 3

**Metareview:**

All reviews see the importance of the motivation of such user study.  And apprecite the  recruiting medical professionals for the user study and provide several very interesting result to compare the five visualization.

However, all reviews also point that the current work is kind of preliminary work for this important data visualization method.  More analysis, a specific visualization should be done for the specific medical data.  And all scores given by the reviewers are near to the boundary of accept and reject.  Based on the current reviews, I prefer to give a boarder line accept.

---

### Decision · Program_Chairs · 2021-01-16

Accept